# Contamination Survey of Insect Genomic and Transcriptomic Data

**DOI:** 10.3390/ani14233432

**Published:** 2024-11-27

**Authors:** Jiali Zhou, Xinrui Zhang, Yujie Wang, Haoxian Liang, Yuhao Yang, Xiaolei Huang, Jun Deng

**Affiliations:** State Key Laboratory of Ecological Pest Control for Fujian and Taiwan Crops, College of Plant Protection, Fujian Agriculture and Forestry University, Fuzhou 350002, China; 15395870850@163.com (J.Z.); xinruiz0387@163.com (X.Z.); 15520727193@163.com (Y.W.); hxleungcanton@163.com (H.L.); y2909255220@163.com (Y.Y.)

**Keywords:** contamination, genomic/transcriptomic database, Insecta, COI barcoding, source

## Abstract

The ignorance of data quality such as data contamination will cause incorrect conclusions and misdirection. Insects are the most diverse group of animals, and the data are increasing rapidly. Although some researchers are aware of the existence of contamination, they mainly detect contamination for individual species and lack systematic evaluation of Insecta data. Here, this study highlights the serious issue of contamination in public databases and emphasizes the importance of verifying data quality before researchers re-use them.

## 1. Introduction

Over the past few decades, the rapid advancement of high-throughput sequencing technologies, driven by platforms like Illumina, Oxford Nanopore, and Pacific Biosciences, has enabled the parallel sequencing of millions to billions of DNA fragments [1], greatly expanding its application in genomics, transcriptomics, metagenomics, and other fields [2]. Thus, it has made the complete sequencing of the human genome a reality in a short time [3]. The explosion of sequencing data has increased significantly due to improvements in efficiency and cost reductions in molecular and sequencing technologies. However, this rapid increase in sequencing data raises concerns regarding contamination, which occurs when sequences from multiple species are present within a dataset [4]. Meanwhile, the establishment of public databases such as GenBank has greatly facilitated data sharing and utilization. The accumulation of sequencing data containing contamination gradually increases when stringent contamination controls are lacking [5].

The ubiquity of microbes and the leading role of humans in experimental operations have made bacteria and human sequences become the common contaminants when sequencing other organisms [6,7]. Although public databases provide massive and open resources, data reliability and accuracy depend on submissions from every laboratory or researcher. The neglect of data quality, such as foreign DNA contamination, may cause a multitude of incorrect conclusions, including misjudgments of horizontal gene transfer [8,9], biases of phylogenomic trees [10,11], and the appearance of spurious protein [12]. For instance, the published mitogenomes of Coccoidea [13] were proved to come from contaminants from their parasitic wasps [14]. Ensuring the high quality of data is crucial, and accessing the contamination within existing databases has become an urgent matter to address. While data contamination in databases has garnered attention, much of the focus has been on prokaryote and human contamination in public databases [6,15,16,17]. Despite being the most abundant group of animals, the evaluation of contamination in Insecta genomic/transcriptomic databases is still lacking, particularly when it comes to studying contaminants between insects or between insects and other animals.

Tools such as KmerID [18], Conterminator [19], KrakenUniq [20], FCS-GX [21], and CLARK [22] can quickly and sensitively recognize different species based on K-mer comparison and strongly rely on whole genome data in reference libraries. It has been shown that only 1737 complete insect genomes are available on NCBI, while there are 994,767 known insect species worldwide on The Catalogue of Life (accessed in August 2024). Thus, assessing the contamination between insect species is challenging when only depending on whole genomes. Using DNA barcodes is a more practical approach, as it allows for the identification of a greater number of species compared to genomic reference databases. The total number of specimens with barcoding exceeds ten million, with over 20,000 species on the Barcode of Life Data System (accessed in August 2024). 16S rRNA is the barcoding for the identification of bacteria [23]. ContEst16S uses it to check for the presence of contamination from prokaryotes by multiple sequence alignment and can find the potential contaminants [24]. In the identification of animal-origin contamination, researchers are more likely to use mitochondrial gene cytochrome c oxidase I (COI), as it shows great ability to distinguish between different species and is widely accepted as a practical and standardized species-level barcoding region [25,26].

MitoGeneExtractor can extract potential COI from the sequences of genomic/transcriptomic databases [27], and RDP classifier can identify these COI sequences by providing confidence scores [28]. In this study, these two tools were used to assess contamination in genomic and transcriptomic data. To compare the levels of contamination between genomic and transcriptomic data across different types of insects, we selected data from four insect orders, including Hemiptera, Coleoptera, Hymenoptera, and Diptera, in the GenBank database. Many Diptera and Hymenoptera species are regarded as parasitic insects, while numerous Hemiptera and Coleoptera species are well-known pests in agriculture and forestry. The following problems were addressed in the current study: (1) the comparison of the difference in contamination between the genomic and transcriptomic databases; (2) the evaluation of the contamination level of four orders and the exploration of the superfamily of insects in each order with the most contamination; and (3) the investigation of the possible causes of contamination.

## 2. Materials and Methods

### 2.1. Initial Data Preparation

The WGS (Whole Genome Sequencing) and TSA (Transcriptome Shotgun Assembly) assemblies of Hemiptera, Hymenoptera, Coleoptera, and Diptera (as detailed in Appendix A) were downloaded from GenBank (https://www.ncbi.nlm.nih.gov/Traces/wgs/, accessed on 8 May 2024). In total, there were 2796 assemblies of WGS and 1382 assemblies of TSA, about 960 GB in size, covering over 1700 species.

### 2.2. Workflow of Investigating Contamination

Based on data from the four orders, this study investigated contamination from different Insecta orders (Insecta-origin contamination) and Mammalia orders (Mammalia-origin contamination). The MitoGeneExtractor (v1.9.5) was used to extract potential COI sequences from these four orders’ data, utilizing COI amino acid references of Insecta and Mammalia (https://github.com/cmayer/MitoGeneExtractor/tree/main/Amino-Acid-references-for-taxonomic-groups/COI-references-for-different-taxonomic-groups, accessed on 15 January 2024). The RDP classifier (v2.13) was used to obtain the identities of these potential COI sequences, including hosts and possible contaminants [29]. The nucleotide collection (nr/nt) database was chosen to run BLAST. The final contaminated sequences were confirmed through two steps, as follows: (1) filtering taxonomic assignments with a strict confidence score of at least of 0.8 using the RDP classifier [30]; and (2) identifying the best top bitscore subject, with threshold values of 70% coverage and 80% identity. All contaminants in the insects and mammals were identified at the order level (Figure 1).

### 2.3. The Analysis of Contaminated Data

The data distribution of four insect orders in WGS and TSA was plotted using Origin 2021 (OriginLab, Northampton, MA, USA). To compare the contamination differences between WGS and TSA, the total number of contaminated assemblies in both types was counted, and a comparative analysis was performed using IBM SPSS Statistics 26 (IBM, New York, NY, USA). The contaminated assemblies were then analyzed to identify which orders and superfamilies were the most affected.

### 2.4. Classification of Contamination Source

To understand the causes of contamination, papers related to the target species and contaminants were searched based on databases such as Web of Science, PubMed, and Google Scholar (accessed on 1 July 2024). Then, we carefully evaluated the potential causes of contamination based on the biological relationships between these organisms. The potential causes of contamination were classified into five major categories, as follows: (1) Food Contamination: this occurs when the original creatures and the contaminants have a predatory relationship; (2) Parasitism Contamination: this involves a parasitic relationship between the original creatures and the contaminants; (3) Collection Contamination: this occurs when the original creatures and the contaminants have similar small sizes, distributions, and habitats; (4) Cross-Contamination: this involves assemblies where contaminants from different projects are mutually included and were submitted by the same authors within the same timeframe or the contamination from human and mouse samples; and (5) Unknown: there is no clear evidence demonstrating a relationship between the original organisms and the contaminants.

## 3. Results

### 3.1. The Overview of Data

The total number of downloaded assemblies was 4178, which included Hemiptera (WGS: 182, TSA: 295), Coleoptera (WGS: 502, TSA: 386), Hymenoptera (WGS: 759, TSA: 418), and Diptera (WGS: 1353, TSA: 283) (Figure 2a). Among these assemblies, we detected a total of 184 cases with contamination, resulting in an overall contamination rate of 4.40%. The contamination rate in the TSA data (11.00%) was significantly higher than that found in the WGS data (1.14%) (*p* = 0.007 < 0.05). The overall contamination degree varied significantly among the four orders, with Hemiptera having the highest overall contamination rate at 9.22%, followed by Hymenoptera at 7.09%, Coleoptera at 3.48%, and Diptera, with the lowest rate, at 1.89%. The highest contamination rate appeared in the TSA data of Hymenoptera, at 13.64%, and the lowest was in Diptera in WGS, at less than 1% (Figure 2b).

### 3.2. Analysis and Assessment of Contamination for Four Orders

There were 164 assemblies detected with Insecta-origin contamination among the four orders, and the contamination rate was 3.93%. In Coleoptera, 16 Insecta-origin orders were detected, representing the highest number of contamination orders. This was followed by Hemiptera, with 12 orders, Hymenoptera with 9 orders, and Diptera with 5 orders (Figure 3a). Lepidoptera, Coleoptera, Hymenoptera, Diptera, and Hemiptera were the five major sources of contamination, as contaminants from these categories accounted for half of all detected contaminants in each order. Hymenoptera was the primary source of contamination in Hemiptera and Diptera, while Diptera was the primary source of contamination in Hymenoptera and Coleoptera (Figure 3b). Compared to Insecta-origin contamination, Mammalia-origin contamination appeared to be less serious, with only 22 assemblies showing contamination with a contamination rate of 0.53%. Hymenoptera and Diptera, which had the highest number of Mammalia-origin contamination orders, with four mammalian orders, were both additionally influenced by Artiodactyla (Figure 3c). Contamination from Primates and Rodentia were detectable in every order (Figure 3d). Also, we observed that some assemblies had multiple contaminants. A total of 23 assemblies contained contaminants from multiple insect orders, and 6 assemblies showed coexisting contamination from insects and mammals.

In Hemiptera, the contaminated assemblies were primarily concentrated in the superfamilies of Fulgoroidea, Coccoidea, and Pentatomoidea. These were predominantly dominated by Cicadellidae (40.0%), Diaspididae (66.7%), and Pentatomidae (50.0%), respectively. In Coleoptera, Scarabaeoidea contamination (11 assemblies) was particularly severe, with the family of Scarabaeidae (72.7%) being the most affected. In Hymenoptera, Apoidea (18 assemblies) showed significantly more contamination than the other superfamilies, with Braconidae (50%) being the dominant family. This was followed by Ichneumonoidea (nine assemblies) and Vespoidea (eight assemblies), which were both from Braconidae and Vespidae, respectively. In Diptera, Muscoidea was prominent, with all contamination being derived from Muscidae (Figure 4).

### 3.3. The Contamination Causes

Food, collection contamination, and cross-contamination could be found in every order (Figure 5). More than half of the contamination causes were unknown, with Coleoptera accounting for the largest proportion of this. In known causes, food contamination was prevalent in Coleoptera, Hymenoptera, and Diptera, especially for the parasitic insects in the last two orders. Hymenoptera experienced the most severe food contamination, with a proportion of 32.76%, which is much higher than the others. Parasitic contamination was the primary cause in Hemiptera, accounting for 17.50%. However, there was no case of parasitic contamination discovered in Coleoptera. For cross-contamination, the proportion was comparatively higher in Diptera than in the other orders, at 10.17%. Collection contamination occurred slightly in the four orders, accounting for a relatively small proportion (Hemiptera: 3.75%, Coleoptera: 4.41%, Hymenoptera: 5.17%, Diptera: 5.08%).

## 4. Discussion

Food contamination and parasitism contamination are primarily caused by the sequencing host itself. Samples could carry some contaminants like symbionts, parasites, and ingested food [31]. The nutritional content of consumed food can be inferred by sequencing the genomes from the midguts of insects [32]. Food contamination is not new, but it is worth noting that many researchers still ignore the contamination from the digestive system when sequencing the target species of our study. For instance, in Hemiptera, we detected contamination from Orthoptera, likely because live crickets were used to feed the water striders, as recorded in the methods and materials [33]. This is not an isolated case, as feeding pea aphids to *Hippodamia convergens* (Coleoptera, Coccinellidae) [34] and rearing soybean aphids (Hemiptera, Aphididae) to *Aphelinus certus* (Hymenoptera, Aphelinidae) have also been documented [35]. Some food contamination came from mammals. For example, contamination from Bovidae (Mammalia) was found in the GIBC01 assembly of *Haematobia irritans* (Diptera, Muscidae), likely due to its blood-sucking behavior. In cases of endoparasitism, food contamination may occur due to parasitic feeding, which is serious in parasitic insects like Hymenoptera and Diptera. Parasitism contamination is also an important factor when the sequenced species carry parasites [36]. Many parasitic wasps could contaminate their hosts by laying their eggs inside of them, while some insects parasitize hosts by attaching to the outer surface of their hosts. In our study, Hymenoptera and Diptera are two main orders associated with parasitism contamination, with hosts often being contaminated by their potential natural enemies, such as Aphelinidae and Conopidae.

Collection contamination could add foreign DNA into the original specimen. To generate sufficient DNA for NGS sequencing, some small organisms require many individuals [37]. Some researchers consider all organisms living in a gall as part of the same colony [38], and this action may overlook the possibility of co-existing species. Additionally, relying on descriptions and images in the literature to identify samples can lead to the misidentification of some similar insects. The occurrence of cross-contamination typically involves two main stages. Before sequencing, environmental contamination can be introduced at any stage of the experimental procedures. It often originates from sources like reagents or sequencing machines that may be contaminated with residual DNA from humans or previous organisms [39]. Experimental devices that process DNA from multiple organisms in batches may introduce genes from other species into their respective DNA libraries during sequencing, leading to foreign species contamination [40]. Ballenghien et al. provided indirect evidence that the sequencing centers significantly contribute to cross-contamination [41]. We noticed the presence of environmental contamination, including that from humans and model organisms such as mice, in the samples from these four orders. Despite our findings, more than half of the contamination sources in our study remain unknown. The high proportion of unknown other contamination primarily arises from our lack of definitive evidence to classify the contamination into the four established categories. Insects represent a complex group with intricate feeding and parasitic relationships, and the existing literature does not cover all possible feeding types and parasitic relationships. As a result, these instances of contamination with unclear relationships may be categorized as unknown. In order to ensure that the classifications of the four main categories of contamination are based on rigorous scientific evidence, the proportion of unknown contamination may be overestimated.

The data from TSA typically exhibit higher contamination rates compared to that of WGS, as shown in studies such as the one on mite contamination [42]. WGS generally provides deeper sequencing coverage than TSA, enabling more detailed insights into genomic variation. Research by Yin et al. demonstrates that deeper sequencing of mitochondrial DNA (mtDNA) is associated with lower cross-contamination levels [43]. In contrast, a lower coverage depth presents challenges for quality control. In high-coverage sequencing, gene sequences from a specific region are read multiple times, allowing for the verification of the correct sequence. This redundancy helps in removing contamination through quality control measures. Furthermore, the cost of genome sequencing is significantly higher than that of transcriptome sequencing [44], leading some researchers to place less emphasis on the quality of the samples used for TSA. Additionally, many researchers collect TSA samples from different individuals and various developmental stages (instars), and with large sample numbers, and they may not strictly verify each individual before pretreatment, increasing the risk of contamination.

In addition to detecting contamination, we also discovered that some assemblies had updated versions with reduced contamination levels. In the updated version of 227 assemblies, after stricter contamination removal measures, the contamination rate decreased from 8.37% to 3.08%, with a total reduction of 5.29%. It suggests that, although researchers may have detected data contamination, it is nearly impossible to completely eliminate contamination through subsequent data processing. To reduce contamination as much as possible, we propose several recommendations. First, sample pre-treatment should be recognized. For non-enteric microorganism studies, performing starvation treatment on the insects could help to empty their guts and minimize contamination from intestinal contents. This applies whether the insects are collected from the field or raised in laboratory captivity [45]. Second, during the sequencing phase, combining the morphological characteristics with DNA barcode validation can provide more accurate species identification [46]. Third, using samples with labeled inline indexes can effectively reduce cross-contamination from the sequencing center [47]. Additionally, before submitting sequencing data to public databases, it is crucial to use contamination removal tools such as FASTQ Screen [48] and DeconSeq [17] or our workflow to rigorously ensure data quality, particularly in addressing common contamination. Lastly, it is advisable to first verify the data quality again when using data from public databases.

## 5. Conclusions

Based on the findings, this study emphasizes the widespread and significant contamination present in WGS and TSA data for Hemiptera, Coleoptera, Hymenoptera, and Diptera within the GenBank database, identified through COI barcoding. By correlating the contaminants and original organisms with existing research, food contamination is revealed as a primary source of contamination in Coleoptera, Hymenoptera, and Diptera, and parasitism contamination is the primary source in Hemiptera. These results offer valuable insights into the common sources and distribution of contamination, highlighting the need for stringent data quality control. This study provides a workflow to detect the potential contamination and causes for future genomic and transcriptomic analysis.

## Figures and Tables

**Figure 1 animals-14-03432-f001:**
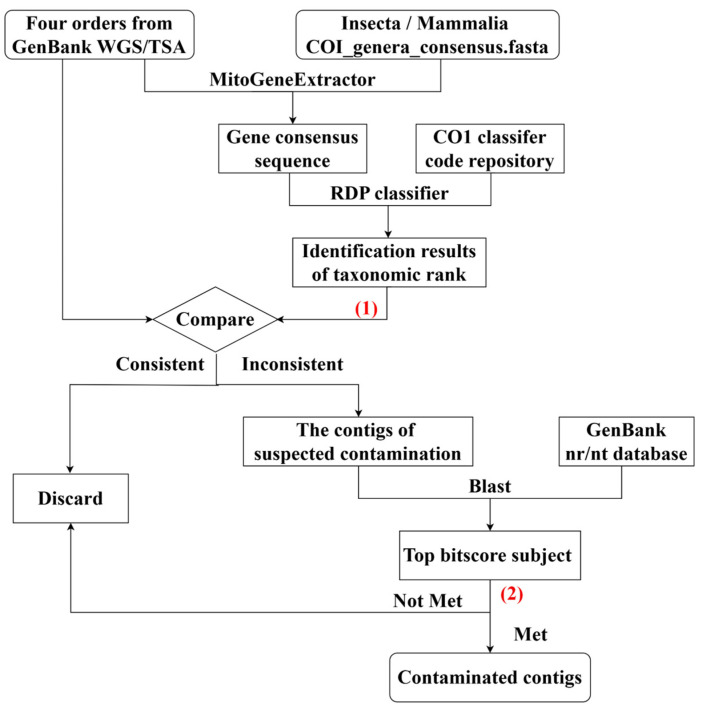
Overview of the workflow used to scan Insecta-origin and Mammalia-origin contamination. Two steps (1–2) for filtering the contamination are marked red. (1) was to filter taxonomic assignments with a strict confidence score of at least 0.8; and (2) was to identify the best top bitscore subject, with threshold values of 70% coverage and 80% identity.

**Figure 2 animals-14-03432-f002:**
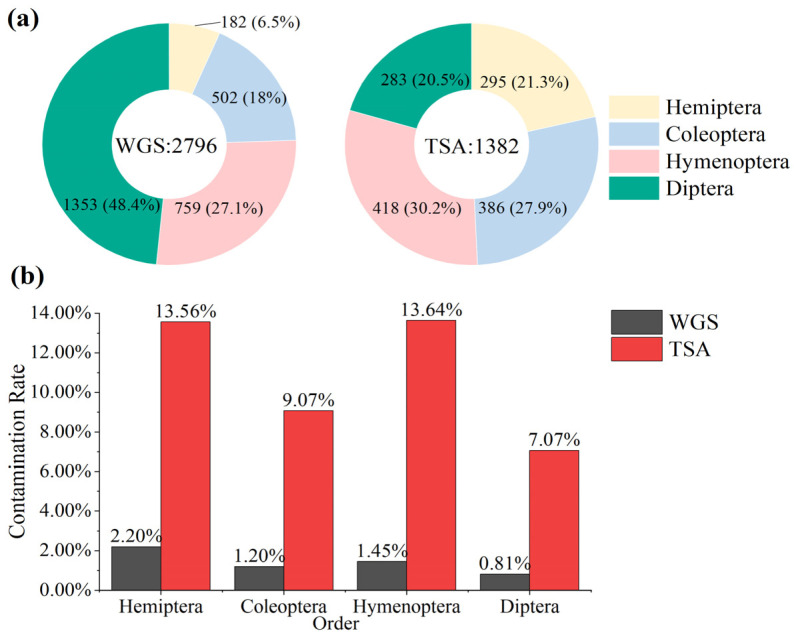
The overview of downloaded data (**a**) and the contamination rates between WGS and TSA in four orders (**b**).

**Figure 3 animals-14-03432-f003:**
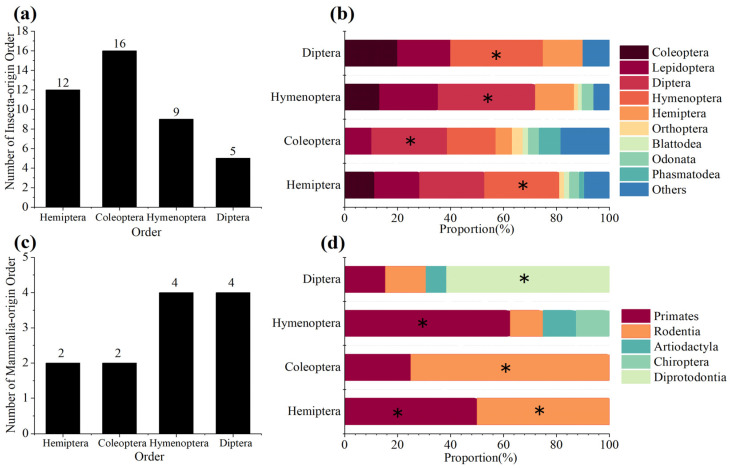
The number of contaminants across different orders within the class Insecta (**a**) and the distribution of Insecta-origin contamination among four insect orders (**b**). The number of contaminants across different orders within the class Mammalia (**c**) and the distribution of Mammalia-origin contamination across four mammalian orders (**d**). The number of assemblies involved in each contamination order as a proportion of the total number of assemblies contaminated in each category. The asterisk (*) indicates the contamination order with the highest proportion in each category.

**Figure 4 animals-14-03432-f004:**
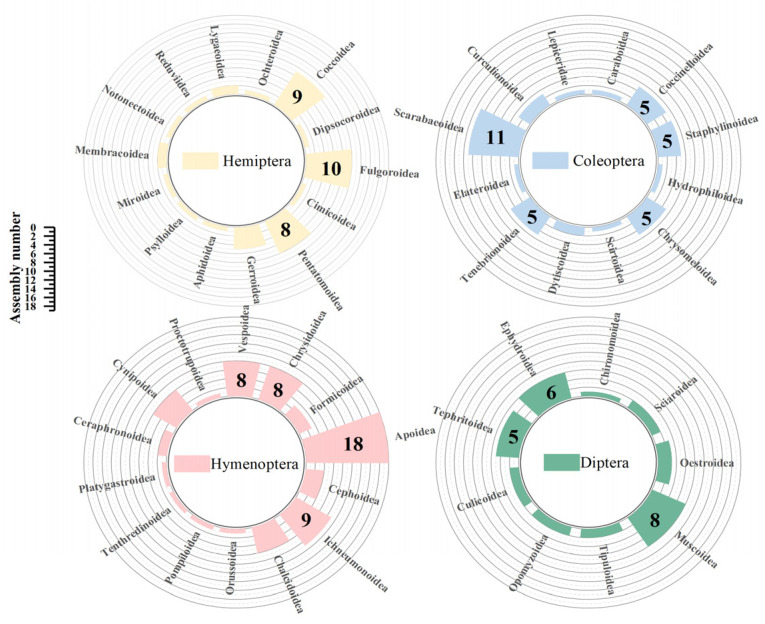
The distribution of contaminated superfamilies in four orders (including Insecta-origin and Mammalia-origin contamination).

**Figure 5 animals-14-03432-f005:**
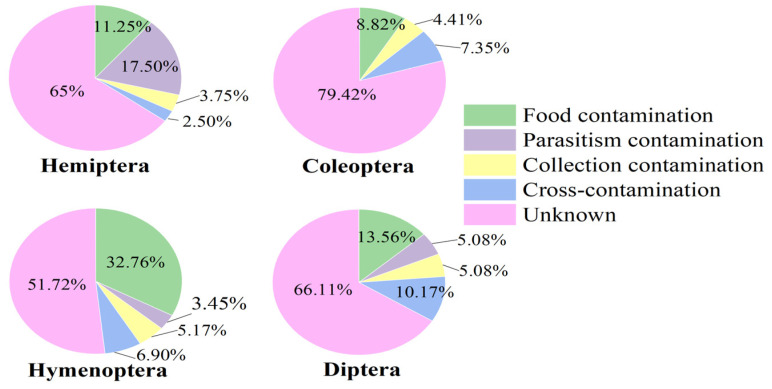
The distribution of contaminated superfamilies in four orders (including Insecta-origin and Mammalia-origin contamination).

## Data Availability

The GenBank WGS/TSA datasets used for this study can be downloaded at [GenBank] https://www.ncbi.nlm.nih.gov/genbank/ (accessed on 8 May 2024). The COI amino acid references of Insecta and Mammalia can be downloaded at https://github.com/cmayer/MitoGeneExtractor/tree/main/Amino-Acid-references-for-taxonomic-groups/COI-references-for-different-taxonomic-groups (accessed on 15 January 2024). The bioinformatic codes for MitoGeneExtractor and RDP classifier are available at https://github.com/cmayer/MitoGeneExtractor (accessed on 15 January 2024) and https://github.com/terrimporter/CO1Classifier (accessed on 15 January 2024).

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
