# Peer review of "Contamination Survey of Insect Genomic and Transcriptomic Data"

_animals, 2024, doi:10.3390/ani14233432_

Round 1

Reviewer 1 Report

Comments and Suggestions for Authors

In the presented work, the authors analyzed whole genome sequences and transcriptomes of four insect orders published in databases by other authors. The objective of the work was to identify contamination in the presented data and try to determine the source of contamination. For this purpose, the available barcoding sequence data were used. Several bioinformatics programs were used to compare the data.

As a result, the authors state different degrees of contamination of whole-genome and transcriptomic data in four orders of insects. As expected, most of the identified contaminants come from the food sources of the objects. This is not news and there are recommendations to overcome such errors. Indeed, when sequencing whole genomes, insects are used together with the contents of their stomachs.

Unfortunately, more than half of the non-specific sequences are of unknown origin. In the diagrams and results, all discrepancies are called combinations. In the discussion, the authors mention that these may be either unknown living organisms or simply errors in evaluating the sequencing results. However, they do not make any assumptions about the possible share of these two types of errors-contaminations. Although, by comparing raw and updated genomic data of some organisms, such an analysis can be carried out and it is possible to assume what percentage of errors is simply due to bioinformatics shortcomings.

The disadvantages of the text include multiple repetitions of the same or similar statements. Especially in the Introduction. But also, in the Results and Discussion. For example, 150-152, 221-224. Figure 2 is not shown.

Recommendations:

conduct a text analysis for repeatability

add an analysis of the proportion of genome assembly errors to the proportion of animal-origin contaminations

Author Response

Comment 1.

In the presented work, the authors analyzed whole genome sequences and transcriptomes of four insect orders published in databases by other authors. The objective of the work was to identify contamination in the presented data and try to determine the source of contamination. For this purpose, the available barcoding sequence data were used. Several bioinformatics programs were used to compare the data.

As a result, the authors state different degrees of contamination of whole-genome and transcriptomic data in four orders of insects. As expected, most of the identified contaminants come from the food sources of the objects. This is not news and there are recommendations to overcome such errors. Indeed, when sequencing whole genomes, insects are used together with the contents of their stomachs.

Response 1:

As the reviewer noted, contamination from food sources is indeed a common issue in insect genome studies. Our research highlights that some researchers still underestimate its impact, often neglecting sample pre-treatments such as starvation protocols or removing digestive tract tissues (lines 204-220). Our study further underscores this problem. Additionally, by analyzing four different insect orders, we found that in Hemiptera, the primary contamination source was not, as expected, from food but rather from parasitism (Figure 5). This result supports the rationale for analyzing contamination sources by insect order. In our discussion section, we also provide specific recommendations for overcoming such errors (Lines 259-276).

Comment 2.

Unfortunately, more than half of the non-specific sequences are of unknown origin. In the diagrams and results, all discrepancies are called combinations. In the discussion, the authors mention that these may be either unknown living organisms or simply errors in evaluating the sequencing results. However, they do not make any assumptions about the possible share of these two types of errors-contaminations.

Response 2:

Thank you for pointing this out. We agree with this comment. We have revised our description of “unknown” to make it easier to understand accordingly in the Materials and Methods section (Lines 132-134) and provided an explanation for the high proportion of unknown origins in the Discussion. “The high proportion of unknown contamination primarily arises from our lack of definitive evidence to classify other contamination into the four established categories of contamination. Insects represent a complex group with intricate feeding and parasitic relationships, and the existing literature does not encompass all possible feeding types and parasitic relationships. As a result, these instances of contamination with unclear relationships may be categorized as unknown. To ensure that the classifications of the four main categories of contamination are based on rigorous scientific evidence, the proportion of unknown contamination may be overestimated”. (Lines 237–244)

Comment 3.

Although, by comparing raw and updated genomic data of some organisms, such an analysis can be carried out and it is possible to assume what percentage of errors is simply due to bioinformatics shortcomings.

Response 3:

The percentage of errors has been calculated in lines 260-262.

Comment 4.

The disadvantages of the text include multiple repetitions of the same or similar statements. Especially in the Introduction. But also, in the Results and Discussion. For example, 150-152, 221-224.

Recommendations:conduct a text analysis for repeatability

Response 4:

We agree with this comment. We have, accordingly, deleted a mass of similar sentences or modified the expression of the corresponding sentences to avoid multiple repetitions. The following line numbers correspond to those in the original version.

The deleted sentences:

Introduction

4.1 Page 1, lines 35-37.

“Second generation sequencing has extensive applications in genomics, transcriptomics, metagenomics and other fields due to its high throughput, high accuracy and low cost”

4.2 Page 2, lines 47-49.

 “In public database, bacteria and human sequences could be easily discovered in the sequencing data of many other organisms [6–8]”

4.3 page 2, lines 54-46.

“Dissemination of low-quality data and wrong conclusion would not only result in wastage of resource but also misdirect subsequent researchers”

4.4 Page 2, lines 67-69.

 “There is no doubt that having standard genomes of contaminants is essential, as aligning high-quality whole genomes would be more accurate for proper identification”

4.5 page 2, lines 75-77.

 “DNA barcoding is more extensive and covers a greater number of species compared to genomic reference databases”

4.6 Page 2, lines 82-84.

 “At the same time, the number of COI sequences is increasing rapidly, providing significant convenience for identifying a diverse range of species.”

Results

4.7 Page 4, lines 152-154.

“Specifically, the contamination rates in TSA data were higher than the overall level (4.40%), while those in WGS data were lower” and As it showed that P=0.007< 0.05, there was a significant difference between WGS and TSA data in contamination rate.”

4.8 Page 5, lines 173-174.

“Although the overall contamination rate in Coleoptera was lower than the average (4.40%),”

4.9  Page 7, line 203

“Food contamination was prevalent across all four orders.”

Discussion

4.10 Page 8, lines 221-222

“In our study, food contamination was a major issue, largely due to the contents of the insects' guts.

4.11 Page 8, line 223-224

As a result, undigested food can act as a contaminant, affecting the sequencing data of predators.”

4.12 Page 8, line 229-230

 resulted in contamination and “introduced their genome into”

4.13 Page 8, 235-236

This was particularly common in the parasitic insects like Hymenoptera and Diptera, where over 60% of food contamination was attributed to this reason.

4.14 Page 9, 267-269

  The primary goal of genome projects is to achieve a contiguous and complete genome assembly [46], which requires high coverage depth to mitigate sequencing error [47].

Comment 5.

Figure 2 is not shown.

Response 5:

Thank you for pointing this out. We have added the detailed cites of this Figure in the result part. (Line 139 and line 147)

Comment 6.

add an analysis of the proportion of genome assembly errors to the proportion of animal-origin contaminations

Response 6:

Thank you for pointing this out. We have added the proportion of insect and animal-origin in line 161.

Reviewer 2 Report

Comments and Suggestions for Authors

The authors of the study submitted aimed to provide evidence for DNA contamination within published genomes from four insect orders as well as discussing possible sources for contaminating DNA. For this purpose, the COI gene barcode sequences deposited in data bases was exploited as diagnostic for unclean DNA.

 In fact, foreign contamination in sequenced genomes has long been suspected but this report represents a piece of a systematic, as well as quantitative view of the problem. For this reason, it can be a useful warning for such a real problem. The manuscript is well written, methodology is clear, good figures illustrate the work done and conclusions are supported by the data presented. I have only a few observations for the authors as follows.

In the Introduction, the authors mention two papers (29,30) to give literature on the usefulness and importance of the COI gene for species discrimination. There are obviously many papers on this subject, but I think it is more adequate in this case to cite not specific papers but instead recent reviews on the issue. If this was the case, keep the references as such. If not, the authors may want to consider replacing by at least one recent review.

Figure 5: Did you mean cross-contamination?

Author Response

Comment 1.

In the Introduction, the authors mention two papers (29,30) to give literature on the usefulness and importance of the COI gene for species discrimination. There are obviously many papers on this subject, but I think it is more adequate in this case to cite not specific papers but instead recent reviews on the issue. If this was the case, keep the references as such. If not, the authors may want to consider replacing by at least one recent review.

Response 1:

Thank you very much for your comments and professional advice. We agree with this comment. Therefore, we have recited a recent review (26) of COI to replace these two specific papers in line 73.

Comment 2.

Figure 5: Did you mean cross-contamination?

Response 2:

Yes, we have corrected the misspelled word in Figure 5. (Page 6)

Reviewer 3 Report

Comments and Suggestions for Authors

Zhuo et al investigated the possible contamination reasons and cases in insect genomic and transcriptomic data. It was an interesting approach and provided great insight into the sequencing aspects. However, there are some major and minor points to consider:

Major

Rather than proceeding with the story by dividing it into four insect orders, it would be better to directly show evidence of the sequencing errors belonging to each of the five categories (Food, Parasitism, Collection, Cross-Contamination and Unknown). There is no visible evidence of the contamination cases on figures and tables. The scientific basis for dividing contamination is unclear.  

Minor:

1.    In materials and methods, if When you run Blast, you need to specify which database you ran it on.

2.    I think the quality of the NGS samples used should be stated in the supplementary material.

3.    It should contain information about the version of MitogENEextractor. Not just this one but all software needs a version.

4.    Can you clarify the “threshold values of 70% coverage and 80% percent identity”?

Figure 1. there is no MARKED RED. Figure quality needs to be improved

Figure 2 and 3. show that the number of libraries used varies by species, but I don't understand what it means to express the number of contaminations by species.

Line 45: needs to be rewritten : The facts that microbes are ubiquitous and humans play a leading role in various experimental operations allow bacteria [6] and human [7] sequences to become the principal and common sources of contamination when sequencing data of other organisms.

Line53: every lab or researcher    lab à Full word

Line 65: KmerID, Conterminator, KrakenUniq, FCS-GX, CLARK  need to put and

Line 103:  In total, there were 2796 assemblies of WGS and 1382 assemblies of TSA with about 960 GB in size, covering over 1,700 species.    Need to put  ,  every thousand. 2,796

Line 132 :  accessed on July 1st   need to put the year as well

Line 151 Firstly, the contamination rate in TSA data was significantly higher than that in WGS data for every order. WGS data were lower.   Other than higher and lower, the author need to put the exact number

Line 176 ;  below 10.  You’d better write as nine orders and five orders.

Line Family - capitalization and genus - italic

Author Response

Comment 1.

Major: Rather than proceeding with the story by dividing it into four insect orders, it would be better to directly show evidence of the sequencing errors belonging to each of the five categories (Food, Parasitism, Collection, Cross-Contamination and Unknown). There is no visible evidence of the contamination cases on figures and tables. The scientific basis for dividing contamination is unclear. 

Response 1:

Thank you very much for your comments and professional advice. Firstly, the reason why we divided the story into four insect orders is that we discovered that the primary cause was different among four orders. Food contamination was prevalent in Coleoptera, Hymenoptera and Diptera. Additionally, we found that in Hemiptera, the primary contamination source was not, as expected, from food but rather from parasitism (lines 187-197). This new finding supports the necessity of analyzing contamination sources by insect order.

Secondly, as for the evidence of contamination causes, we have listed several cases in Discussion (Lines 201-244). The specific results of the contaminant BLAST comparisons are listed in the Spreadsheet S2.

The criteria for classifying contamination types are presented in the Methods section, and we have revised the definition of contamination to enhance clarity. (Lines 133-134).

Comment2.

Minor:

2.1 In materials and methods, if When you run Blast, you need to specify which database you ran it on.

Response 2.1:

Thank you for pointing this out. We agree with this comment. Therefore, we have shown specify database “the Nucleotide collection (nt/nr) database” in lines 102-103.

2.2 I think the quality of the NGS samples used should be stated in the supplementary material.

Response 2.2:

Thank you for pointing this out. We agree with this comment. It’s a pity that there is no Busco value of every assembly in WGS/ TSA assemblies. Therefore, we have added some information (contigs total length and contigs count) of every assembly in Supplementary Files (Spreadsheet S1) and this may help know the basic quality of NGS samples.

2.3 It should contain information about the version of MitogENEextractor. Not just this one but all software needs a version.

Response 2.3:

Thank you for pointing this out. We agree with this comment. Therefore, we have added the version of MitoGeneExtractor and RDP classifier in line 97 and line 101.

2.4 Can you clarify the “threshold values of 70% coverage and 80% percent identity”?

Response 2.4:

 In this study entitled that Diversity and Distribution of Mites (ACARI) Revealed by Contamination Survey in Public Genomic Databases, an 80% threshold has been applied for filtering, and we adopted this in our research. However, we encountered some contigs with lengths reaching thousands of base pairs. To avoid discarding these longer contigs with slightly lower coverage, we applied a 70% coverage threshold and kept an 80% Per Ident.

In fact, most of our results are significantly above this threshold (Spreadsheet S2), indicating that our findings are reliable.

2.5 Figure 1. there is no MARKED RED. Figure quality needs to be improved

Response 2.5:

Thank you for pointing this out. We agree with this comment. Therefore, we have changed the color of step (1) and (2) in this figure (Page 3).

2.6 Figure 2 and 3. show that the number of libraries used varies by species, but I don't understand what it means to express the number of contaminations by species.

Response 2.6:

We have revised the description of the ordinate in Figure 3 (a and c) to make it easier to understand (Page 5).

2.7 Line 45: needs to be rewritten : The facts that microbes are ubiquitous and humans play a leading role in various experimental operations allow bacteria [6] and human [7] sequences to become the principal and common sources of contamination when sequencing data of other organisms.

Response 2.7:

Thank you for pointing this out. We agree with this comment. Therefore, we have rewritten this sentence in lines 43-45.

2.8 Line53: every lab or researcher    lab à Full word

Response 2.8:

Thank you for pointing this out. We agree with this comment. Therefore, we have used laboratory to replace lab in line 46.

2.9  KmerID, Conterminator, KrakenUniq, FCS-GX, CLARK  need to put and

Response 2.9:

Thank you for pointing this out. We agree with this comment. Therefore, we have put the word of “and” in line 59.

2.10 Line103, In total, there were 2796 assemblies of WGS and 1382 assemblies of TSA with about 960 GB in size, covering over 1,700 species.    Need to put  ,  every thousand. 2,796

Response 2.10:

Thank you for pointing this out. We agree with this comment. Therefore, we have put the “,” when using the number of 2796 and 1382 in lines 92-93.

2.11: Line 132:  accessed on July 1st   need to put the year as well

Response 2.11:

Thank you for pointing this out. We agree with this comment. Therefore, we have put the year in line 123.

2.12: Line 151 Firstly, the contamination rate in TSA data was significantly higher than that in WGS data for every order. WGS data were lower.   Other than higher and lower, the author need to put the exact number

Response 2.12:

Thank you for pointing this out. We agree with this comment. Therefore, we have put the exact number of every order in WGS and TSA in lines 141-142.

2.13: Line 176 ;  below 10.  You’d better write as nine orders and five orders.

Response 2.13:

Thank you for pointing this out. We agree with this comment. Therefore, we have changed the number of  9 and 5 in line 154.

2.14: Line Family - capitalization and genus - italic

Response 2.14:

Thank you for pointing this out. We agree with this comment. Therefore, we have modified the format of genus and target species in the Supplementary files (Spreadsheet S1 and S2).